bioinformatics/health and disease and epidemiology/computational biology

machine learning, big data, obesity, monozygotic twins

**Author for correspondence:**
Milla Kibble
e-mail: mmk60@cam.ac.uk

# An integrative machine learning approach to discovering multi-level molecular mechanisms of obesity using data from monozygotic twin pairs

Milla Kibble[1,3], Suleiman A. Khan[1],
Muhammad Ammad-ud-din[1], Sailalitha Bollepalli[1],
Teemu Palviainen[1], Jaakko Kaprio[1,2], Kirsi H. Pietiläinen[4]
and Miina Ollikainen[1]

[1]Institute for Molecular Medicine Finland (FIMM), and [2]Department of Public Health, University of Helsinki, Helsinki, Finland
[3]Department of Applied Mathematics and Theoretical Physics, University of Cambridge, Cambridge, UK
[4]Obesity Research Unit, Helsinki University Central Hospital and University of Helsinki, Helsinki, Finland

MK, 0000-0003-1130-4010; SAK, 0000-0002-0823-4042;
SB, 0000-0002-8773-7149; TP, 0000-0002-7847-8384;
JK, 0000-0002-3716-2455; KHP, 0000-0002-8522-1288;
MO, 0000-0003-3661-7400

We combined clinical, cytokine, genomic, methylation and dietary data from 43 young adult monozygotic twin pairs (aged 22–36 years, 53% female), where 25 of the twin pairs were substantially weight discordant (delta body mass index > 3 kg m$^{-2}$). These measurements were originally taken as part of the TwinFat study, a substudy of The Finnish Twin Cohort study. These five large multivariate datasets (comprising 42, 71, 1587, 1605 and 63 variables, respectively) were jointly analysed using an integrative machine learning method called group factor analysis (GFA) to offer new hypotheses into the multi-molecular-level interactions associated with the development of obesity. New potential links between cytokines and weight gain are identified, as well as associations between dietary, inflammatory and epigenetic factors. This encouraging case study aims to enthuse the research community to boldly attempt new

machine learning approaches which have the potential to yield novel and unintuitive hypotheses. The source code of the GFA method is publically available as the R package GFA.

# 1. Introduction

Worldwide, obesity has nearly tripled since 1975, according to the World Health Organisation. In 2016, more than 1.9 billion adults were overweight, of whom over 650 million were obese [1]. A raised body mass index (BMI), which is a common measure used to define obesity, is a major risk factor for non-communicable diseases such as cardiovascular diseases, type 2 diabetes mellitus (T2DM), chronic kidney disease, musculoskeletal disorders (especially osteoarthritis) and certain cancers [2–4]. There is evidence that high BMI is really driving the unfavourable changes in disease-associated biomarkers [5]. Recently, Romieu *et al.* [6] reviewed the evidence of the associations between energy balance and obesity and concluded that the main driver of weight gain is energy intake that exceeds its expenditure. So excessive energy intake which is not compensated by energy expenditure leads to excess weight gain, which over time can lead to a multitude of chronic health problems.

However, the weight gain trajectory is far from being this straightforward and the determinants of excess weight gain are still not well understood. Key players probably include genetics, epigenetics, type and quality of diet, exercise and lifestyle, microbiome composition and hormonal effects as well as medications acting on the individual. In addition, there are societal and psychological factors acting on populations and groups of persons. The person-level and macro-level factors interact in such a complex manner that often researchers can focus on only one or two of these features at a time [7]. For example, there are studies highlighting the heritability of obesity [8,9], estimates of which range between 40 and 70%. Indeed monozygotic (MZ) twins are very rarely BMI discordant, i.e. differ significantly in weight. The first genetic locus to show robust association with BMI and obesity risk, namely *FTO*, was discovered via a genome-wide association study (GWAS) in 2007 [10,11] and since then, over 500 genetic loci have been found, among other things providing strong support for a role of the central nervous system in obesity susceptibility [8]. It is now clear that most cases of obesity are polygenic and multifactorial and the full picture of how genetic and other factors jointly influence at a molecular level individual preferences for and responses to diet and physical activity remains largely beyond our comprehension.

The wealth of data being collected in this field holds great potential to offer some answers, but the challenge remains how to consider the large and diverse datasets in an integrated manner to infer such multifactorial molecular mechanisms. In addition, ideally one would want to undertake an unbiased study by including all the available clinical and genomic features and then using a systematic and data-driven approach to learn which features are relevant for discovering molecular mechanisms of obesity.

In this paper, we present a machine learning approach that can look at multiple diverse datasets simultaneously and learn associations between the variables in the multiple different datasets in a data-driven manner. In particular, we combine genetic, methylation, clinical, cytokine, dietary and lifestyle data. Rather than simply looking at this data from a set of randomly chosen individuals, we go a step further in order to elucidate the mechanisms of weight *gain* and use data from MZ twin pairs, many of whom are substantially weight discordant—a rare dataset. Thus, we can focus on *differences* in weight that cannot be attributable to genetics alone and can discover how these differences are associated with other factors, such as DNA methylation or diet. The outcome results encouragingly highlight many known associations as well as suggesting novel links that could offer new hypotheses of molecular mechanisms.

We start with a brief background to twin studies to highlight the novelty of our method in this domain. We then aim to describe the method in accessible terms for an interdisciplinary audience and only then proceed to the key results on obesity. Finally, we offer our thoughts on future possibilities of such an approach.

## 1.1. Twin studies

So-called classical twin methods have focused on estimating the heritability of different phenotypes by comparing occurrences of the phenotype in those MZ and dizygotic (DZ) twin pairs where at least one twin exhibits the phenotype of interest. The basic premise is that if genetics influences a particular phenotype, then the occurrence of that phenotype for both twins will be more common within MZ twin pairs, who share their whole genomic sequence, than DZ twin pairs, who share roughly 50% of their segregating genes akin to non-twin siblings [12]. Such heritability studies using large twin repositories cover the whole range of complex phenotypes [13,14] including obesity [8] and many obesity-related

phenotypes. For example, by looking at 1126 twin pairs, Goodrich *et al.* [15] identified heritable bacterial taxa of the gut microbiome. The most heritable taxa were the family Christensenellaceae (phylum Firmicutes) which is enriched in lean individuals and has been shown to limit adiposity gain when given as faecal transplants to mice deficient in the taxa, suggesting that heritable microbes could influence adiposity.

Another popular twin design is the so-called co-twin control method whereby within-pair comparisons of trait discordant MZ twin pairs are used to identify factors associated with the trait, against a background of equivalent age, sex and genotype (which are perfectly matched for the twins) and in part also ruling out the influence of environment (which is partially matched, and termed as shared environment) [16]. The first such studies helped prove the effect of smoking on lung cancer [17]. Illustrative examples relevant to obesity would be the recent study in BMI-discordant MZ twin pairs revealing sub-types of obesity based on both clinical traits and gene expression in subcutaneous adipose tissue [18], as well as that based on DNA methylation in leucocytes [19]. Such studies usually employ classical machine learning approaches such as linear mixed-effects models.

For a thorough exposition of twin studies, the reader is referred to the review article of van Dongen *et al.* [13] and for a discussion of future directions, to the review article of Baird *et al.* [20]. In the current article, we also look at MZ twin pairs discordant for BMI, and in particular use machine learning methods to search for associations in any *differences* between the heavier and leaner individuals in such pairs. Such associations including dietary and lifestyle data could give hints as to behavioural mechanisms for the twins differing in BMI despite sharing the same genotype, whereas associations involving the other types of data could highlight consequences and/or causes of obesity which are genotype independent. To our knowledge, machine learning techniques of the form proposed here have not previously been applied as part of twin studies.

## 1.2. Machine learning

Machine learning and artificial intelligence (AI) have been applied in the field of medicine for over a decade and, with the now routine collection of large datasets in all research fields and the affordability of computing power, the number of application areas is rising sharply. Although to a lesser extent than in the drug discovery sector, machine learning is also being used in innovative ways in the area of disease prevention (see the discussion in [21]) and it is here that our current contribution sits.

Machine learning refers to mathematical algorithms that have been coded into computer programs. Their function is to look at the dataset of interest and 'learn' patterns in that data and possibly also predict data values not available to the researcher. Most algorithms achieve this by having an underlying mathematical model to describe the type of data defined by a set of unknown parameters and then calculating the parameters that best fit the particular dataset. Camacho *et al.* have recently written a very clear introduction to machine learning in biological applications [22], including definitions of the most commonly used terms and examples of recent developments.

In this study, we use an advanced machine learning method called group factor analysis (GFA); see the Material and Methods section. The objective of our study is to discover multi-level molecular mechanisms of obesity. Here, we hypothesize that relationships between clinical, genomic and molecular features provide a proxy to understanding these complex mechanisms and so we model statistical dependencies between genomic, methylation, clinical, cytokine, dietary and lifestyle datasets using GFA. Unlike classical methods, like principal component analysis (PCA) and clustering which are suitable for analysing a single data source only, GFA learns and identifies relationships between multiple data sources. GFA achieves this, by learning latent variables, also known as components, that are shared between two or more datasets. The latent variables are the parameters learned by the model that capture correlated and common patterns between the datasets.

GFA takes as its input multiple datasets (or matrices), where each matrix has samples in its rows and variables in its columns. In our case, the five datasets are each represented as a matrix, one each for clinical, cytokine, genomic, methylation and dietary data. Crucially, for the understanding of the approach used here, our samples correspond to twin pairs and the values in the matrices correspond to the *difference* in value of a variable between the twins in the pair (we always subtract the value for the leaner twin from the value for the heavier twin). So a given row $i$ in the matrices contains our data for a given twin pair and a given column $j$ contains the data for a given variable, such as low density lipoprotein (LDL) cholesterol for example. An entry $ij$ in the matrix would then correspond to the LDL cholesterol value for the heavier twin in the given pair minus the LDL cholesterol value for the leaner twin in the pair. GFA then learns associations between datasets as well as between

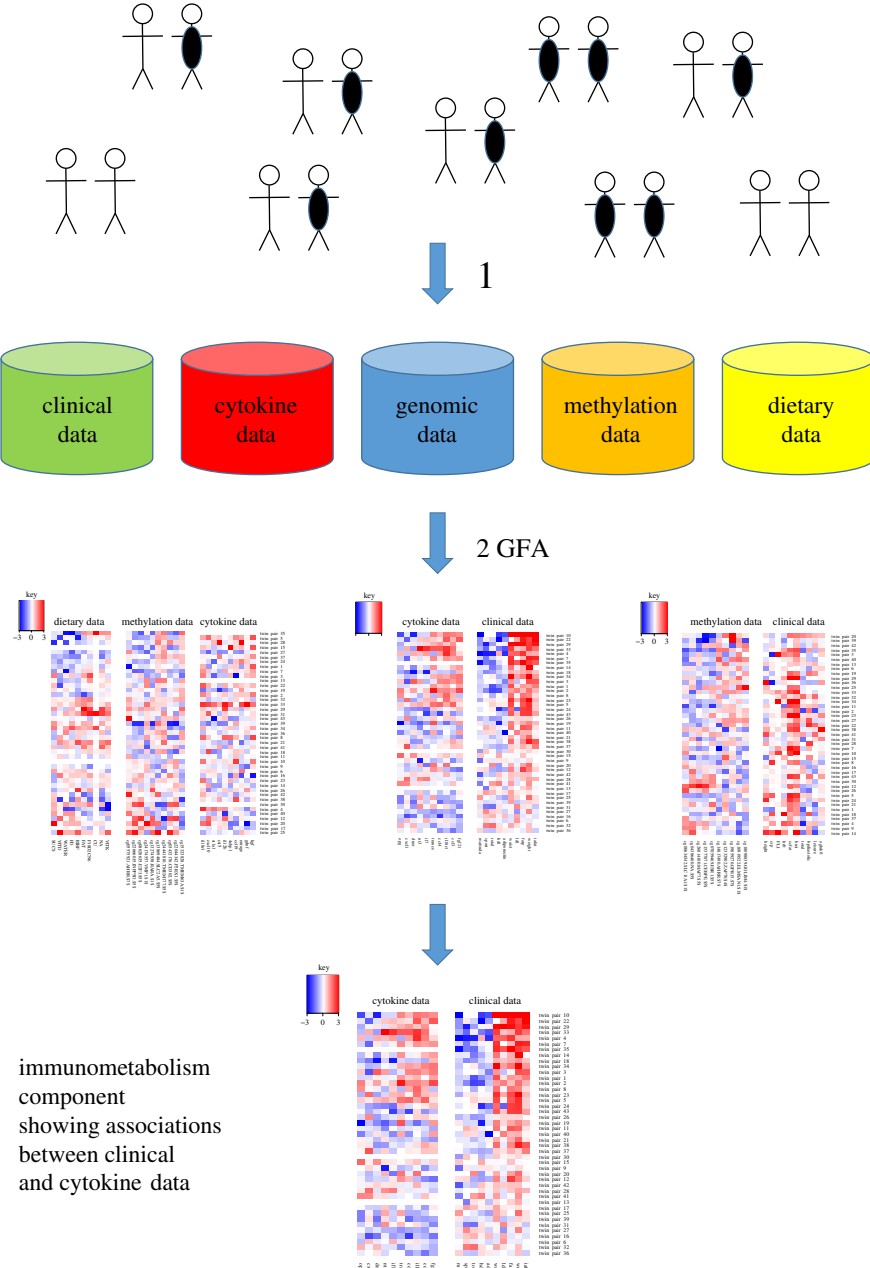

**Figure 1.** Pipeline for the GFA analysis on MZ twin pairs. (1) Clinical, cytokine, genomic, methylation and dietary data were collected from 43 young adult monozygotic twin pairs, where 25 of the twin pairs were substantially weight discordant (delta BMI $> 3$ kg m$^{-2}$). For each twin pair and each variable, the value for the leaner twin was subtracted from the value for the heavier twin. This resulted in five large data matrices comprising 42, 71, 1587, 1605 and 63 variables, respectively. (2) All five large data matrices were input into the group factor analysis (GFA) computational tool, giving rise to 38 so-called component diagrams (three of which are shown in this figure). Each component diagram has up to five small heatmaps picturing the associations discovered within or between the five datasets. The magnified component is the immunometabolism component, also in figure 3.

variables within each dataset, finding associations of the form 'heavier twins who consume a lot more of nutrient $x$ than their leaner twin, tend to have higher levels of cytokine $y$ than their leaner twin and greater methylation at cytosine-phosphate-guanine (CpG) sites $z$ and $w$'.

The output of GFA is distinct components, each representing interpretable relationships across one or more datasets. The relationships can be visualized and interpreted via so-called component diagrams. Each component diagram has up to five small heatmaps picturing the associations discovered within or between the five datasets; figure 1. GFA automatically identifies all the statistical relationships in the data, representing each as a separate component.

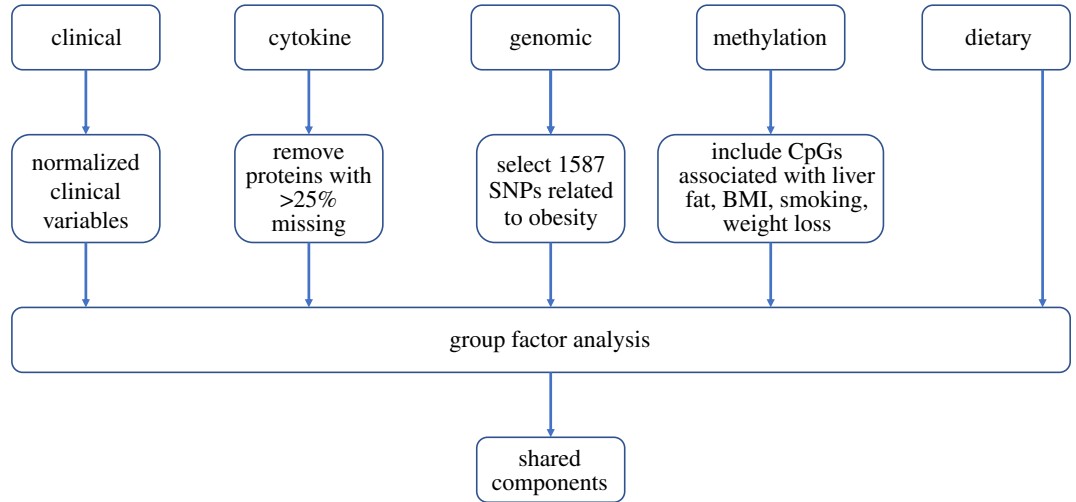

**Figure 2.** A data processing flow-chart for the GFA analysis.

Such novel approaches that integrate multiple sources of information provide an invaluable way to discover connections at multiple molecular levels from vast amounts of data [23–25] or to predict outcomes, as in the landmark study of Zeevi and colleagues where blood parameters, dietary habits, anthropometrics, physical activity and gut microbiota were used to predict personalized postprandial glycaemic response to real-life meals [26]. So far, GFA has only been taken up by the computer science community to develop the methodology further [27,28] and, although in the literature there have been small case studies, few real applications have been previously published. These have, to our knowledge, focused on the problem of elucidating the mechanisms of action of small molecules and so the samples within the matrices have corresponded to experimental and structural information about small molecules [29].

# 2. Material and methods

We combined clinical, cytokine, genomic, methylation and dietary data from 43 young adult MZ twin pairs (aged 22–36 years, 53% female), where 25 of the twin pairs were substantially weight discordant (delta BMI > 3 kg m$^{-2}$). A data processing flow-chart is given in figure 2.

## 2.1. Finnish twin cohort

TwinFat, which is a substudy of The Finnish Twin Cohort study (FTC) [30], is designed to study obesity using an MZ co-twin control design [31,32]. The twin pairs in TwinFat were selected from two population-based twin cohorts, FinnTwin16 and FinnTwin12, comprising 10 full birth cohorts of Finnish twins. Twins were included in the current study based on the availability of data for both members of a pair. All twins were free of somatic and psychiatric diseases and with a stable weight for at least three months prior to the current study. Venous blood samples were drawn in the morning after an overnight fast. Zygosity was confirmed by genotyping. All twins provided written informed consent. The protocols of the FTC TwinFat data collections were approved by the Ethics Committee of the Helsinki University Central Hospital.

## 2.2. Clinical data

Included in the analysis were 42 clinical variables (table 1). Some categorical variables (e.g. gender) are common to both twins and others (e.g. smoking status) may or may not be. For variables such as gender which are common to both twins, we include in the matrix the information from either twin (rather than difference values, which would always be zero).

Gender was labelled as 0 = male, 1 = female. Hence in the component diagrams, red represents the female twins. Twenty twin pairs were male and 23 were female. Smoking status was recorded as never, former or current. Hence Smoking.Current is 1 if the twin currently smokes and 0 otherwise. Smoking.Current.diff is the value of Smoking.Current for the heavier twin minus that for the leaner twin

**Table 1.** Clinical characteristics of the 43 twin pairs in our study.

| variable | full description | unit | min | max | range | mean | variance | min within pair difference | max within pair difference |
|---|---|---|---|---|---|---|---|---|---|
| adiponectin | fasting plasma adiponectin concentration | ng ml⁻¹ | 1236.55 | 9576.3 | 8339.75 | 3172.58 | 236015.58 | 60 | 5644.57 |
| age | age at sampling | year | 22.78 | 36.1 | 13.32 | 30.62 | 15.7 | NA | NA |
| alat | fasting plasma alanine aminotransferase | U/l | 10 | 103 | 93 | 28.93 | 281.95 | 0 | 62 |
| asat | fasting plasma aspartate aminotransferase | U/l | 14 | 63 | 49 | 26.69 | 67.28 | 0 | 34 |
| bmi | body mass index | kg m⁻² | 19.68 | 48.58 | 28.9 | 27.5 | 28.82 | 0.06 | 10.22 |
| bpdiastolic | diastolic blood pressure | mmHg | 48 | 96 | 48 | 72.22 | 90.53 | 0 | 22 |
| bpsystolic | systolic blood pressure | mmHg | 96 | 158 | 62 | 125.11 | 193.81 | 0 | 34 |
| crp | C-reactive protein | mg l⁻¹ | 1.33 | 132.76 | 131.43 | 19.62 | 668.13 | 0 | 72 |
| diabetes | presence of diabetes | 0 = no, 1 = yes | 0 | 2 | 2 | 0.02 | 0.05 | 0 | 2 |
| fatkg | fat mass | kg | 7.73 | 69.71 | 61.98 | 27.53 | 145.13 | 0.08 | 22.77 |
| fatp | percentage body fat | % | 10.3 | 56.3 | 46 | 33.52 | 96.29 | 0.1 | 19.2 |
| ffmkg | fat free mass | kg | 31.67 | 73.08 | 41.41 | 50.83 | 111.81 | 0.01 | 9.76 |
| FLI | fatty liver index [33] | | 0.03 | 51.54 | 51.51 | 3.85 | 94.78 | 0 | 44.09 |
| gender | sex | male = 0, female = 1 | 0 | 1 | 1 | 0.53 | 0.25 | NA | NA |
| gluk0 | fasting plasma glucose concentration | mmol l⁻¹ | 4 | 9.8 | 5.8 | 5.27 | 0.49 | 0 | 4.2 |
| gt | fasting plasma gamma glutamyl transferase concentration | U/l | 10 | 195 | 185 | 28.69 | 1066.63 | 0 | 107 |
| hdl | fasting plasma high density lipoprotein concentration | mmol l⁻¹ | 0.52 | 3.04 | 2.52 | 1.46 | 0.21 | 0 | 0.97 |
| height | height | cm | 152 | 196.5 | 44.5 | 171.26 | 102.11 | 0 | 6 |
| homa | HOMA index | | 0.35 | 7.94 | 7.59 | 1.47 | 1.36 | 0 | 6 |
| hr0 | heart rate | beats min⁻¹ | 44 | 110 | 66 | 70.94 | 124.15 | 0 | 66 |
| iafat | intra-abdominal fat volume | cm³ | 95 | 5878 | 5783 | 1170.69 | 103096.35 | 23.11 | 2186.49 |

(*Continued.*)

**Table 1.** (*Continued.*)

| variable | full description | unit | min | max | range | mean | variance | min within pair difference | max within pair difference |
|---|---|---|---|---|---|---|---|---|---|
| igt | impaired glucose tolerance | | 0 | 1 | 1 | 0.15 | 0.13 | 0 | 1 |
| insu0 | fasting plasma insulin concentration | mIU l$^{-1}$ | 1.4 | 28.8 | 27.4 | 6.23 | 19.27 | 0 | 21.3 |
| col | fasting plasma total cholesterol | mmol l$^{-1}$ | 2.7 | 6.9 | 4.2 | 4.52 | 0.76 | 0 | 2.7 |
| ldl | fasting plasma low density lipoprotein concentration | mmol l$^{-1}$ | 0.98 | 5.1 | 4.12 | 2.71 | 0.67 | 0 | 2.8 |
| leisure | leisure time index [34] | | 1.5 | 4 | 2.5 | 2.81 | 0.34 | 0 | 1.25 |
| leptin | fasting plasma leptin concentration | pg ml$^{-1}$ | 467.58 | 77640 | 77172.42 | 19330.95 | 362561870.3 | 510 | 38361.3 |
| LFS | liver fat score [35] | | -3.71 | 3.41 | 7.12 | -1.67 | 1.58 | 0.04 | 4.01 |
| liver fat | liver fat percentage | % | 0.14 | 24.25 | 24.11 | 2.94 | 21.09 | 0.01 | 15.38 |
| liver.fat.%.prediction | liver fat percentage prediction [35] | | 0.65 | 13.93 | 13.28 | 2.82 | 5.74 | 0.02 | 10.41 |
| liverfat.disc | discordant for liver fat | 0 = no, 1 = yes | 0 | 1 | 1 | 0.48 | 0.26 | NA | NA |
| matsuda | Matsuda index [36] | | 0.84 | 22.71 | 21.87 | 8.64 | 25.28 | 0.06 | 17.78 |
| scfat | subcutaneous fat volume | cm$^3$ | 1072.56 | 11734.1 | 10661.54 | 4208.71 | 435562.18 | 31.8 | 3809 |
| smoking.current | current smoker | 0 = no, 1 = yes | 0 | 1 | 1 | 0.31 | 0.22 | 0 | 1 |
| smoking.former | former smoker | 0 = no, 1 = yes | 0 | 1 | 1 | 0.3 | 0.21 | 0 | 1 |
| smoking.never | smoking never | 0 = no, 1 = yes | 0 | 1 | 1 | 0.38 | 0.24 | 0 | 1 |
| sport | sport index | | 1 | 5 | 4 | 2.73 | 1.06 | 0 | 3 |
| tg | fasting plasma triglycerides | mmol l$^{-1}$ | 0.37 | 4.37 | 4 | 1.1 | 0.43 | 0 | 2.2 |
| total | total physical activity index | | 4.62 | 11.62 | 7 | 8.28 | 2.59 | 0.12 | 3.75 |
| waist | waist circumference | cm | 65.2 | 123 | 57.8 | 89.76 | 166.33 | 0.1 | 32 |
| weight | body weight | kg | 48.7 | 127.8 | 79.1 | 80.87 | 300.13 | 0.6 | 29.1 |
| work | work physical activity index | | 1.5 | 4.88 | 3.38 | 2.74 | 0.68 | 0 | 2 |
| year | study year | | 2006 | 2013 | 7 | 2009.51 | 5.05 | NA | NA |

(hence if the current smoking status for both twins is the same, Smoking.Current.diff will be zero). Age, Gender, Year and Liverfat Discordance were common to both twins and are input directly into the data.

We have calculated liver fat scores (LFS), and liver fat percentage prediction [35], and fatty liver index (FLI) [33], even though we actually have liver fat values measured by magnetic resonance imaging technique. The idea behind this was to see what these scores are associated with. However, we did not observe any interesting associations with liver fat or its related scores.

We also include information on physical activity (leisure time, work, sport and total physical activity indices from the Baecke questionnaire [34]), because higher physical activity level is known to be associated with lower adiposity and better metabolic health, independent of genetics [37].

Most of the twins in the study were metabolically healthy, and did not differ significantly for LDL, Homa index or insulin levels within pairs.

## 2.3. Cytokines

Included in the analysis were 71 cytokines from the Proseek Multiplex Inflammation I panel (Olink Bioscience, Uppsala, Sweden). For statistical comparison, proteins with missing data frequencies above 25% were excluded, leaving 71 proteins out of the original 92 for analysis. Protein levels are presented as normalized protein expression values following an inter-plate control normalization procedure.

## 2.4. Genotype information

Chip genotyping was done using Illumina Human610-Quad v.1.0 B, Human670-QuadCustom v.1.0 A and Illumina HumanCoreExome (12 v.1.0 B, 12 v.1.1 A, 24 v.1.0 A, 24 v.1.1 A) arrays. The algorithm for genotype calling was Illumina's GenCall for all HumanCoreExome chip genotypes and Illuminus for 610 k and 670 k chip genotypes. Genotype quality control was done in two batches (batch1: 610 k + 670 k chip and batch2: HumanCoreExome chip genotypes), removing variants with call rate below 97.5% (batch1) or 95% (batch2), removing samples with call rate below 98% (batch1) or 95% (batch2), removing variants with minor allele frequency below 1% and Hardy-Weinberg Equilibrium $p$-value lower than $1 \times 10^{-06}$. Also samples from both batches with heterozygosity test method-of-moments $F$ coefficient estimate value below 0.03 or higher than 0.05 were removed along with the samples which failed sex check or were among the multidimensional scaling PCA outliers. The total amount of genotyped autosomal variants after quality control were 475 637 (batch1) and 221 814 (batch2). We then performed pre-phasing using EAGLE v.2.3 [38] and imputation with MINIMAC3 v.2.0.1 using the University of Michigan Imputation Server [39]. Genotypes of both batches were imputed to the 1000 Genomes Phase III reference panel [40].

Included in the analysis were 1587 single nucleotide polymorphisms (SNPs) and we use the information from either twin (rather than the difference values), as both twins in a pair have almost exactly the same genotype. Somatic DNA mutation does occur with age [41] and will lead to minor variation even between MZ twins. MZ twins with large BMI discordance are extremely rare. When there is discordance, it usually arises around the age of 16–20 but many BMI discordant pairs do not continue to be discordant when followed over time [5]. This emphasizes the importance of genetic influences on weight regulation. Therefore, it is important to find the triggers to obesity and mechanisms involved for those that have a genetic predisposition.

To focus our analysis on the most important findings, we chose to include in the analyses SNPs associated with obesity and obesity-related traits. An additional motivation for this choice is that, as we found in previous work [29], it is difficult to draw meaningful or actionable hypotheses from genes for which nothing is known. The main article used to choose the SNPs for this analysis was the GWAS meta-analysis of Locke *et al.* [8]. We also used SNPs retrieved from searches for BMI, liver disease, metabolic syndrome and diabetes from the NHGRI-EBI GWAS Catalogue [42] as well as the SNPs from the paper of Turcot *et al.* [43] on rare variants associated with BMI (electronic supplementary material, table S1; the number in between dollar signs refers to the source from which the SNP was chosen, and is also included in the component diagrams). SNPs that are present in more than 38 pairs were removed as these are unlikely to be linked with the differences in the pairs and would only bias the model towards a different 'locally' optimal solution. For each SNP, we assigned the values of no risk allele 0, one risk allele 1 and two risk alleles 2.

## 2.5. Methylation information

DNA methylation has been shown to be both stable and dynamic. Across the human postnatal lifetime, stability in methylation is primarily owing to genetic contributions, while environmental exposures

contribute to methylation dynamics [44]. Twin studies have shown that DNA methylation profiles are more divergent in older twins than infant twins, and although stochasticity may have a role in this phenomenon, these findings also add support to the influence of environmental factors to the epigenome [45].

Several epigenome-wide association studies have now been conducted and have identified a panel of gene loci where methylation levels significantly differ in obese and lean individuals. We include in our analysis DNA methylation data measured on Illumina's Infinium HumanMethylation450 BeadChip. DNA methylation measured as beta values (ranging from 0 to 1) was preprocessed using the R package methylumi [46] and normalized by beta-mixture quantile normalization [47]. We did not adjust for cell-type composition specifically because we wanted to contain all clinical variation that associates with the obesity phenotype, including very low-grade inflammation, which may result in differences in the cell type compositions within the BMI discordant twin pairs. The ComBat function in the R package SVA [48] was used to correct for potential batch effects. Including all CpGs in our analysis would have resulted in high-dimensional matrices and introduced a large number of noisy sites in the modelling process. Therefore, to focus the analysis on the most significant findings, DNA methylation values of a set of pre-selected 1605 CpGs were input into the GFA method. CpG sites were selected from the recent meta-analysis of Wahl *et al.* [49]. They show that BMI is associated with widespread changes in DNA methylation and genetic association analyses demonstrate that the alterations in DNA methylation are predominantly the consequence of adiposity, rather than the cause. We also include CpGs associated with elevated liver fat [19], CpGs whose methylation has been previously shown to differ in the adipose tissue of BMI-discordant MZ twin pairs [50], smoking-associated CpGs [51], and CpG sites that have been associated with weight loss [52] (electronic supplementary material, table S2).

## 2.6. Dietary data

Included in the analysis were 63 dietary variables (electronic supplementary material, table S3). Total energy and macronutrient intake were assessed with 3-day food records. Subjects were given clear oral and written instructions by a registered dietician on how to keep the food record (two working days and one non-working day) and they were encouraged to keep their usual eating patterns and to estimate the amounts of all foods and drinks using household measures. The conversion of data from the records into nutrient values was performed by a dietician using the program DIET32, which incorporates the national Finnish database for food composition (Fineli$^R$). The nutritional composition of new ready-prepared meals that were not included in the DIET32 program was obtained from the manufacturers. Daily energy intake is expressed in kilocalories (kcal) and macronutrient intakes are expressed as percentages of total energy intake. Results are presented as the mean ± s.d. of the 3 days [53]. Information on habitual diet was estimated using a qualitative food-frequency questionnaire incorporating 52 food and non-alcoholic beverage items that are common in the Finnish diet [54].

## 2.7. Group factor analysis

GFA is formulated as a method to identify statistical dependencies between multiple datasets. The method learns a joint integrated model of the datasets with matched samples (i.e. having a common set of samples), to extract meaningful and interpretable information [27,55,56]. GFA has been successfully used for identifying structural properties predictive of drug responses [24], cross-organism toxicogenomics [57] as well as highly accurate drug response predictions [28].

Specifically, GFA takes a set of matrices $\mathbf{X}^{(m)}$, where $m = 1 \ldots M$, and identifies patterns of statistical dependencies across all of them. The model learns these statistical dependencies in a data-driven fashion, automatically identifying the type and amount of dependencies. Therefore, GFA learns a low-dimensional space $\mathbf{Z}$ (of $K$ components) that represent the variation patterns across all datasets. A component can be active in one or more datasets, meaning that it captures the relationships between those particular datasets. This is achieved by modelling the total variation of all of the datasets while inducing structured sparsity. This characteristic allows GFA to automatically identify dependency patterns that are shared across any subset of the datasets, in a truly data-driven fashion.

Formally, GFA is defined in a Bayesian formulation for datasets $\mathbf{X}^{(m)} \in \mathcal{R}^{N \times D_m}$, where $m = 1 \ldots M$, having $N$ matched samples and $D_m$ dimensions, as a product of the Gaussian latent variables

$\mathbf{Z} \in \mathcal{R}^{N \times K}$ and projection weights $\mathbf{W}^{(m)} \in \mathcal{R}^{D_m \times K}$

$$\mathbf{x}_n^{(m)} \sim \mathcal{N}\left( \mathbf{W}^{(m)} \mathbf{z}_n , \sum{}^{(m)} \right),$$
$$\mathbf{z}_n \sim \mathcal{N}(0, \mathbf{I})$$
$$\mathbf{w}_{d,k}^{(m)} \sim h_{m,k} \mathcal{N}(0, (\alpha_{d,k}^{(m)})^{-1}) + (1 - h_{m,k})\delta_0$$
$$h_{m,k} \sim \text{Bernoulli}(\pi_k)$$
$$\pi_k \sim \text{beta}(a^\pi, b^\pi)$$
$$\alpha_{d,k}^{(m)} \sim \text{gamma}(a^\alpha, b^\alpha).$$

Here, $\sum^{(m)}$ is a diagonal noise covariance matrix. GFA induces component-wise sparsity on the projections $\mathbf{w}_{:,k}^{(m)}$ using a beta-Bernoulli distribution. As a result, the projection weights $\mathbf{w}_{:,k}^{(m)}$ for a component $k$ are active for the one or more datasets ($m$), capturing dependencies specific to one dataset or shared between datasets.

The R package GFA v.1.0.3 [58], is used to model the five datasets, dietary, clinical, genomic, methylation and cytokine. In the genomic data, SNPs with values in three or less samples and more than 38 samples were removed as outliers. In the clinical dataset, the categorical variable smoking (never, former, current) was one-hot encoded to form binary representation. Additionally, all variables with more than 50% of values missing were left out of the matrices. As a result, for the $N = 43$ twin pair samples, the clinical data contained $D_1 = 42$ variables, cytokine $D_2 = 71$, genotype $D_3 = 1587$, methylation $D_4 = 1605$ and dietary $D_5 = 63$. All datasets except genotype were scaled to unit variance.

The model, implemented using Gibbs sampling, was run with $K = 40$ components which was deemed large enough owing to the presence of empty components, as recommended by Virtanen *et al.* [55]. In order to model the large amount of noise in the data, we used an informative noise prior with *a priori* noise variation set to one-third [27]. Finally, a total of 2000 sampling iterations were run, with the first 90% corresponding to burn-in while the last 10% representing the posterior.

GFA found 38 components to model the five datasets. Components active in two or more datasets, such as clinical and cytokine, represent variation common to the two datasets. Components specific to a dataset, for example, genotype, represent variation patterns that are consistent within the genotype but not correlated with other datasets. Both types of components represent interesting relationships in our case and are examined in this study.

# 3. Results

## 3.1. Group factor analysis applied to data from monozygotic twin pairs

In our analysis, samples refer to twin pairs and the data for each twin pair corresponds to *difference* values for each of the variables (i.e. the value of the variable for the heavier twin minus the value for the leaner twin of each pair). The aim of the analysis is to identify drivers or consequences of *increase* in weight, which are difficult to distinguish by this study design. The genetic data suggest (see the end of this section) that in fact most of the 43 twin pairs in our data have a genetic predisposition to obesity. Hence the analysis has the potential to elucidate why some individuals are faring better than others, despite their genetic burden. This could be invaluable knowledge for informing prevention strategies.

The output of GFA applied here is 38 sets of associations between variables within and between the different datasets. As mentioned above, each such set of associations between variables is called a component and can be visualized as up to five adjacent heatmaps corresponding to the five datasets, showing how the relevant variables in each dataset are associated with each other. To ensure accurate interpretation of the component heatmaps, we point out that positive values are in red and negative values in blue. Therefore, a red square in the heatmap would indicate that for the given twin pair, the heavier twin has a higher value of the variable than the leaner twin. Hence the column for BMI, for example, is red in colour for all twin pairs.

With the analysis producing 38 components, it is not possible here to go through all components in detail. Instead, we pick six interesting examples, choosing to include some accompanying component diagrams in the supplementary files rather than in the main body of text. We stress that many components highlight known associations, thus adding to the credibility of this approach, and we point these out for those components discussed.

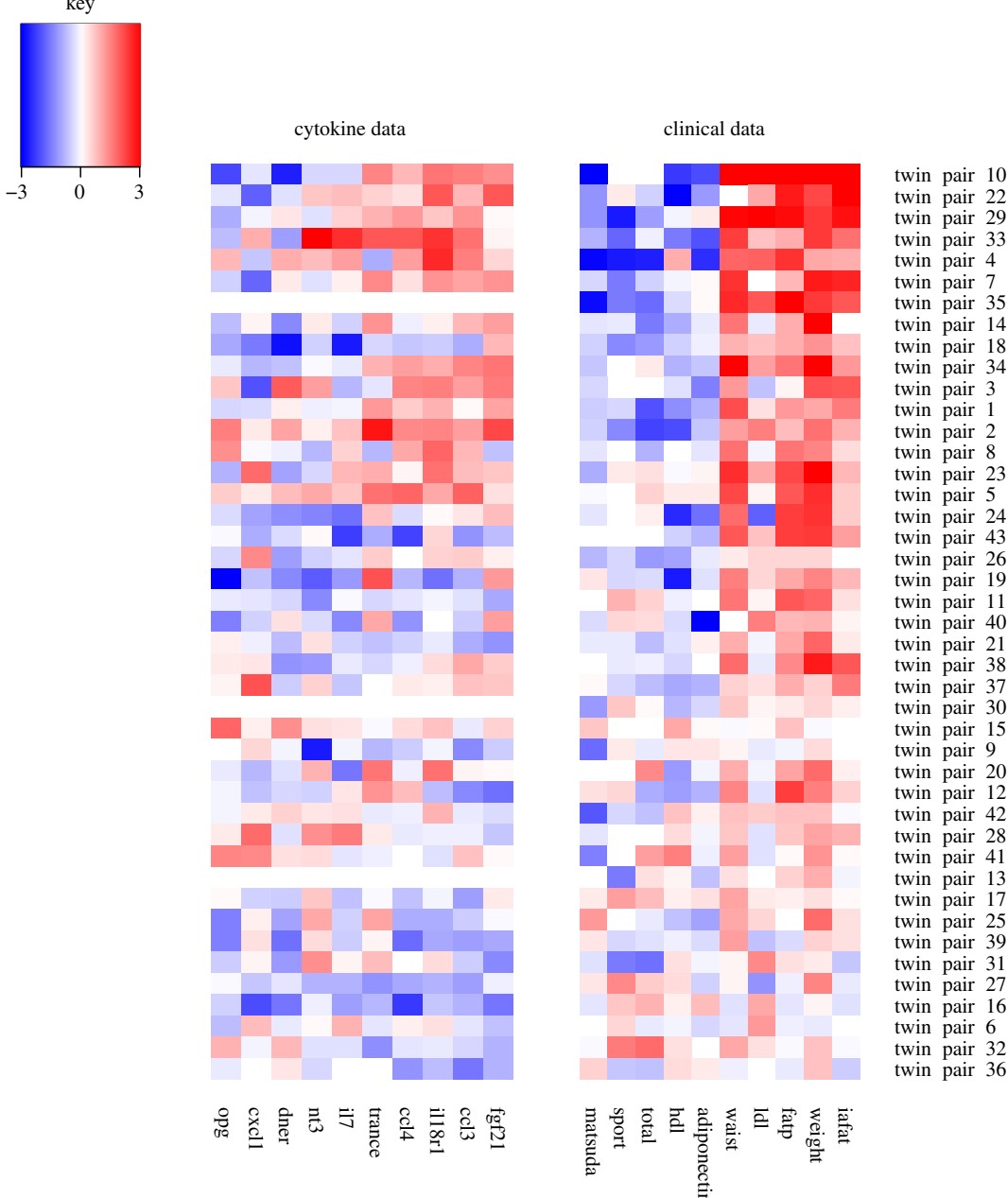

**Figure 3.** The immunometabolism component. The method has picked up associations between clinical data and cytokine data. The twin pairs seem to be ordered roughly by fat percentage (fatp) discordance, with the most discordant pairs at the top of the picture also having high negative HDL and adiponectin difference (in other words the heavier twin in the pair has a lower value of HDL and adiponectin than the leaner twin). Because we are working with difference values (value for the heavier twin in the pair minus the value for the leaner twin in the pair), the BMI column will throughout the analyses be completely red (indicating positive values), because the BMI for the heavier twin minus the BMI for the leaner twin will always be by design a positive value. Likewise, through all of the components certain other variables highly correlated with BMI are completely red, such as weight, subcutaneous fat, liver fat and fat percentage, and variables inversely correlated with BMI are blue, such as adiponectin. Full description of the clinical variables: adiponectin, fasting plasma adiponectin concentration; fatp, percentage body fat; hdl, fasting plasma high density lipoprotein concentration; iafat, intra-abdominal fat volume; ldl, fasting plasma low density lipoprotein concentration; matsuda, Matsuda index; sport, sport index; total, total physical activity index; waist, waist circumference; weight, body weight.

In what we have called the immunometabolism component (figure 3), the GFA method has picked up associations between two of the five datasets, namely the clinical data and the cytokine data. These results are completely data driven and the associations of the clinical variables with obesity are all well-known [59,60]. For the twins that differ most within pair with respect to adiposity (weight, waist, fat percentage and intra-abdominal fat) as well as LDL cholesterol (seen at the top of the figure in red),

the heavier twin tends to have much lower insulin sensitivity (depicted by the Matsuda index [36]), less physical activity (sport and total indices from the Baecke questionnaire [34]), high density lipoprotein (HDL) and adiponectin (shown by the blue colour). What may yield novel insights are the links with the cytokine data. For the twin pairs with the above profile, the heavier twin tends to have elevated values of the cytokines TRANCE, CCL4, IL18R1, CCL3 and FGF21. It has been known since the early 2000s that many immune mediators are abnormally produced or regulated in obesity, contributing to altered metabolic status [61].

For example, the method clearly picks up fibroblast growth factor 21 (FGF21), a hormone that is known to have elevated levels in insulin-resistant morbidities such as obesity and T2DM [62]. Elevated FGF21 levels in these diseases are suspected to be signs of FGF21 resistance [63], similar to the concept of insulin resistance. FGF21 is a peptide hormone secreted by multiple tissues, most notably the liver. Indeed, we have previously shown that high values of FGF21 (measured by the enzyme-linked immunosorbent assay; ELISA) are associated with high liver fat [64]. Elevated levels also correlate with liver fat content in non-alcoholic fatty liver disease [65]. Significantly increased FGF21 levels in circulation have been detected in patients with muscle-manifesting mitochondrial diseases, in which case most of the circulating FGF21 arises from the muscle [66]. Interestingly, an SNP of *FGF21*—the rs838133 variant—has been identified as a genetic mechanism responsible for the sweet tooth behavioural phenotype, a trait associated with cravings for sweets and high sugar consumption [67,68].

The other four cytokines included in this component also have some previous links to obesity, highlighting the potential of the method to offer both known and novel hypotheses on the mechanisms of immunometabolism. TRANCE (RANKL) has been proposed to link Metabolic Syndrome and osteoporosis [69]. The CC chemokine family members CCL3 and CCL4 have both tens of publications where obesity is mentioned, but with debate on the mechanisms involved and function [70]. Finally, although there is not much reference to IL18R1 and obesity, its ligand, IL18, has in several studies been associated with obesity, insulin resistance, hypertension and dyslipidemia (see [71] and the references therein).

A second component, the HDL component, has also picked up the clinical and cytokine datasets, but this time the twin pairs seem to be ordered roughly by HDL discordance rather than weight discordance, with the twin pairs at the top of the heatmap picture (electronic supplementary material, figure S1) being those for whom the heavier twin has lower levels of HDL compared to the leaner twin. This associates with the heavier twin having higher levels of cytokines CCL11, UPA, FGF19, TRANCE and SCF, again offering some potentially novel associations. Not much is known about these cytokines in relation to obesity or HDL, apart from for FGF19 which, like its related hormone FGF21 mentioned earlier, is being investigated as a pharmacological target for obesity [72]. In contrast to FGF21 however, metabolic diseases exhibit reduced serum FGF19 levels [73]. The simultaneous increase in serum FGF21 levels is probably a compensatory response to reduced FGF19 levels, and the two proteins concertedly maintain metabolic homeostasis [74]. It is interesting then that here levels of FGF19 are increased when weight gain is accompanied by a large reduction in HDL levels. It is also interesting to note that though fat percentage difference is present in this component, its values are not exactly correlated with HDL difference.

Next, we consider a component where within twin pair differences in clinical variables have been associated with methylation differences. In the leisure time physical activity component, leisure time physical activity [34] seems to be associated with methylation changes independently of BMI difference (figure 4). When the heavier twin partakes in less leisure activity, they also have higher levels of methylation at CpG sites on *SLC11A1*, *MAP7*, *CEBPE* and *ESR1* and lower levels of methylation at *AHRR*, *ZAP70*, *GPR15*, *ELMSAN1* and *GOLIM4*. We observe that for most substantially weight discordant pairs, the twin with greater leisure time activity is leaner, and we have shown elsewhere that the more active twin, even in MZ pairs, remains leaner [75–77].

It is highly challenging to link nutrient and immune responses, let alone combining this with epigenetic alterations. Here, we offer a contribution in this area. We highlight the results from three components, all of which picked up associations with dietary variables.

In the epigenetic component (figure 5), we see associations between the dietary, methylation and cytokine data. When there is a clear lower consumption in the heavier twin of sucrose, vitamin D, water, fluoride and riboflavine, then there is a lower methylation of CpGs at *AHRR*, *INPP5D*, *E2F3*, *VMP1* and *RARA* and higher values of cytokines 4EBP1, CCL19, ENRAGE, GDNF and HGF. Although crude, this gives a glimpse into the ability of the method to highlight possible connections between these three data levels, where difference in diet is not influenced by genetics and differences

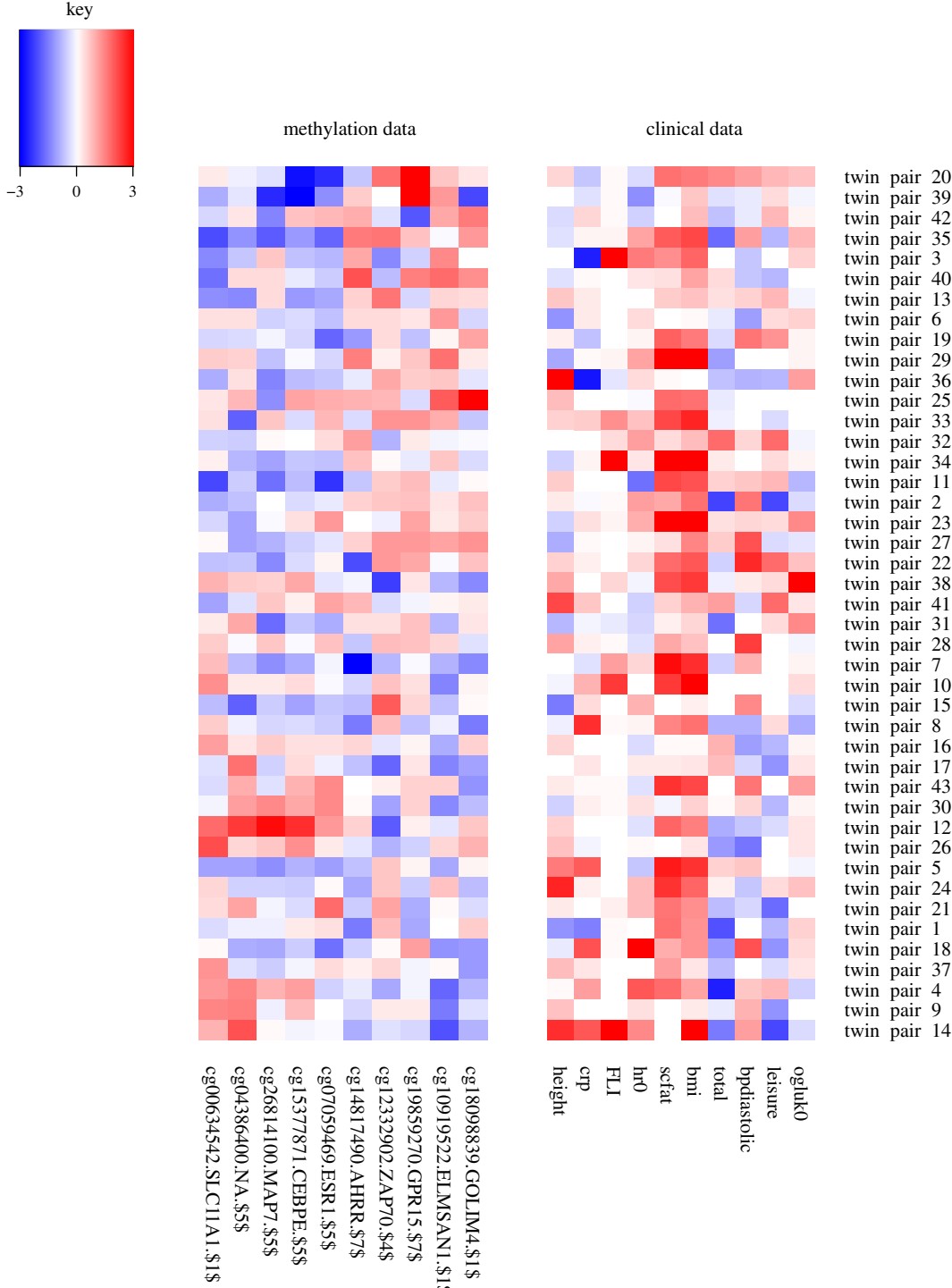

**Figure 4.** The leisure time physical activity component. As can be seen in table 1, the maximum within pair difference in the leisure time activity index is 1.25, which can be considered large (these index values usually have mean 3 and a range of 3.25). Full description of the clinical variables: bpdiastolic, diastolic blood pressure; bmi, body mass index; crp, C-reactive protein; FLI, fatty liver index; height, height; hr0, heart rate; leisure, leisure time index; ogluk0, fasting plasma glucose concentration; scfat, subcutaneous fat volume; total, total physical activity index.

in consumption are associated with differences in molecular features. All of the above genes have been linked to cytokines (mainly interleukins) and inflammation in the literature and also all have associated studies on miRNAs, but it is difficult to draw any conclusions without further experiments. For the cytokines, the fit to the literature is difficult to decipher: 4EBP1 has been shown to have a protective effect in obesity in male mice [78], ENRAGE has been shown to be positively correlated with visceral fat adiposity [79], and elevated HGF levels induced by a high-fat diet have been shown to have a

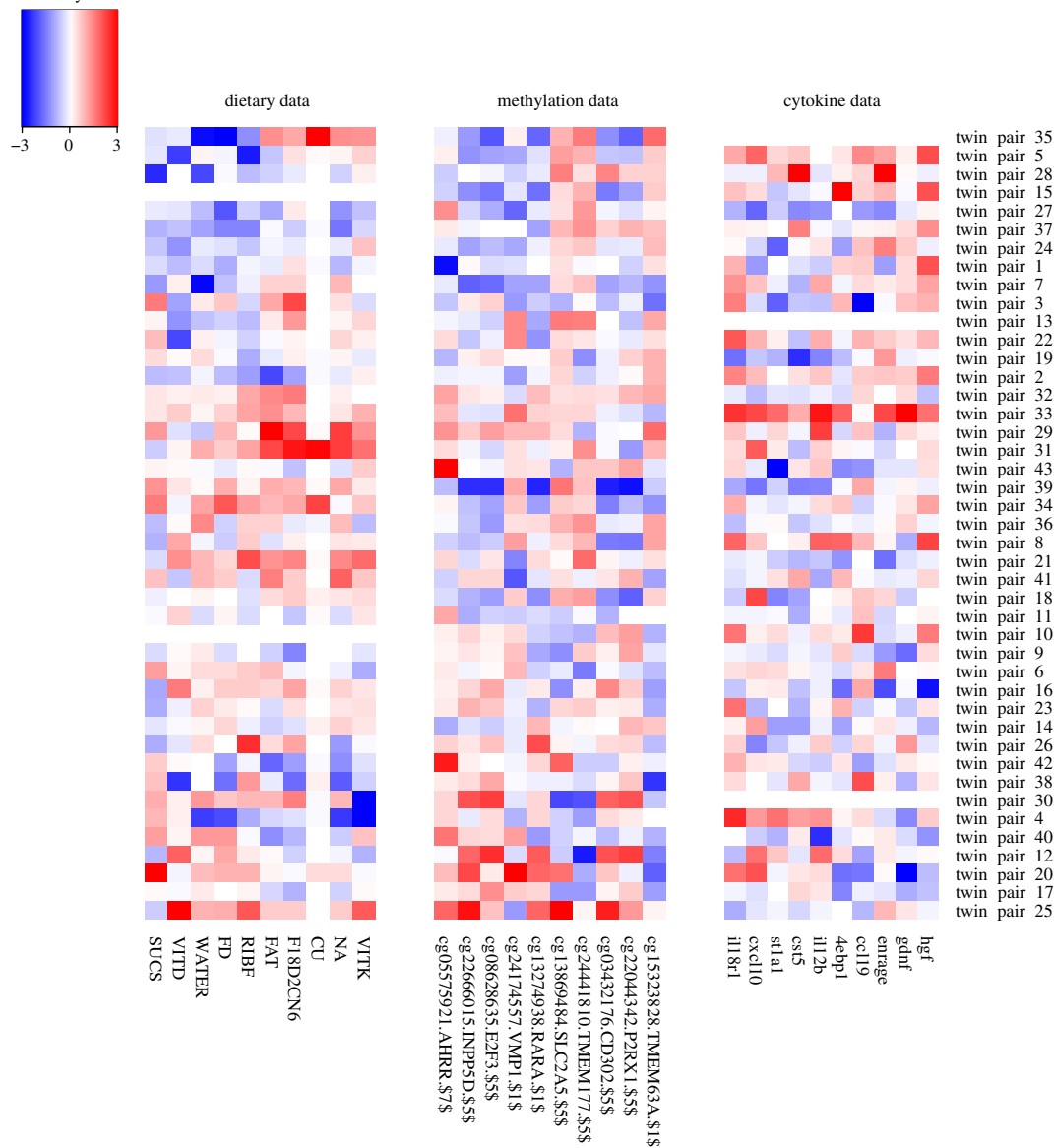

**Figure 5.** The epigenetic component. Full description of the dietary variables: CU, copper; FAT, fat; F18D2CN6, fatty acid 18:2-n6; FD, fluoride; NA, sodium; RIBF, riboflavin; SUCS, sucrose; VITD, vitamin D; VITK, vitamin K; WATER, water.

protective role against obesity and insulin resistance [80]. GDNF family members have also been implicated in obesity [81].

We find further links between diet and methylation in the sucrose component (electronic supplementary material, figure S2). For the twin pairs at the top of the picture, the heavier twin consumes more sucrose and has increased methylation in CpGs at *CD3E*, *SDF4*, *FAM53B* and *AHRR* (a different *AHRR* CpG site to that in the epigenetics component) and decreased methylation in CpGs at *LY6G6F*, *C1orf127*, *IRX1*, *PPIAP3* and *MARCH11*. These individuals also consumed more carotene, vitamin C, sugar, copper and less vitamin D, molybdenum, selenium, lactose and N3 polyunsaturated fatty acids than the leaner twin in the pair. There is an apparent contradiction here between high levels of sugar and, for example, vitamin C, but it is possible that this sugar is being consumed from fruit.

It is particularly interesting that *AHRR* appears for a third time in our analysis. The methylation of this gene is commonly associated with smoking [82–84], and we do mildly see this *AHRR*—smoking association in one of our components (results not shown; the task of finding associations with smoking is hindered by the difficulty in coding into the method differences in past and current smoking status between the twins). However, there are also some previous links between *AHRR* and weight, for example, in a study showing that offspring DNA methylation of *AHRR* is associated with

maternal BMI and birth weight [85]. Because two of our GFA components associate *AHRR* methylation with sucrose intake, we could hypothesize that sugar may also contribute to the methylation seen at *AHRR*; nutritional stimuli have been shown to contribute to *AHRR* expression [86]. There is a well-established link between *AHRR/AHR* and inflammation, and dietary flavonoids and indolens, tryptophan and arachidonic acid [87–89].

In the starch component (electronic supplementary material, figure S3), we have weak links between diet and clinical variables and cytokines, but do retrieve known links between lower starch consumption and lower levels of adiponectin [90]. We see clearly that individuals within twin pairs who tend to eat more starch, also tend to eat more fatty acid 18 : 3-n3, polyunsaturated fatty acids, fluoride and carbohydrate. This may indicate use of starchy vegetables, and an overall healthier fat consumption as well. This is the only component including the gender variable, and even here the association looks quite weak. It would have been interesting to see some associations which only occurred in males or females, but we did not observe this in the current study.

Finally, the component diagrams can offer insights into the individuals in the analysis which can help inform the hypotheses drawn. With regards to the varying genetic load of the twin pairs, all appear to be susceptible to weight gain. Electronic supplementary material, figure S4 shows five genes for which almost all of the twins have at least one risk allele, as well as interestingly another five genes associated with obesity for which hardly any of the twins have a risk allele. However, the electronic supplementary material, figure S5 shows that the twin pairs divide into roughly two sets based on their genetic burden at five SNPs at three genes, including *FTO*. Polygenic risk scores (PRS) for obesity have been available for some time [91], with recent efforts producing a good prediction of those individuals at a high risk for obesity [92]. In this work, we decided to retain SNP level information in order to facilitate the elucidation of molecular interactions at the genetic and other levels. However, in future work, inclusion of a suitably weighted continuous PRS value could potentially distinguish differing mechanisms at high versus low PRS for obesity.

## 3.2. Results summary

The method has found many known clinical characteristics of obesity in a data-driven manner. Beyond this validation of the method, more encouraging are the suggested links between key obesity-related features and mechanisms of immunometabolism. It is well known that obesity is associated with changes in the production of hormones, adipokines and cytokines [61,93–95]. A review of the growth of the field of immunometabolism and latest developments is given in two recent papers by Hotamisligil [61,93]. The work here also addresses a key question stated by Hotamisligil as to whether mechanisms can be identified that integrate nutrient and immune responses [93].

Although in the current work we are only observing associations, and so cannot claim anything regarding causality, the associations can suggest hypotheses to investigate, such as that sucrose or other dietary factors and inflammation affect methylation at *AHRR*. This could ultimately also add to efforts to develop better biomarkers of nutritional intake. We saw in the immunometabolism component that obesity and known related clinical variables associated with elevated levels of cytokines TRANCE, CCL4, IL18R1, CCL3 and FGF21, for which known links to obesity were discussed. In the HDL component, HDL was associated with cytokines CCL11, UPA, FGF19, TRANCE and SCF. The leisure time physical activity component, we see that when the heavier twin partakes in less leisure time physical activity, they also have higher levels of methylation at CpG sites on *SLC11A1*, *MAP7*, *CEBPE* and *ESR1* and lower levels of methylation at *AHRR*, *ZAP70*, *GPR15*, *ELMSAN1* and *GOLIM4*.

Three further components focused on links between nutrient intake, immune response and methylation. We see in the epigenetic component that when there is a clear lower consumption in the heavier twin of sucrose, vitamin D, water, fluoride and riboflavine, then there is a lower methylation of CpGs at *AHRR*, *INPP5D*, *E2F3*, *VMP1* and *RARA* and higher values of cytokines 4EBP1, CCL19, ENRAGE, GDNF and HGF. In the sucrose component, we see that when the heavier twin consumes more sucrose, carotene, vitamin C, sugar, copper and less vitamin D, molybdenum, selenium, lactose and N3 polyunsaturated fatty acids than the leaner twin in the pair, then they have increased methylation in CpGs at *CD3E*, *SDF4*, *FAM53B* and *AHRR* and decreased methylation in CpGs at *LY6G6F*, *C1orf127*, *IRX1*, *PPIAP3* and *MARCH11*. Finally, the starch component highlights the known association between lower starch consumption and lower levels of adiponectin.

Given the design of our analysis, looking at difference values within MZ twin pairs genetically predisposed to weight gain but where some individuals are faring better than others, the associations

found have the potential to give hints as to why some twins of a pair differ in BMI despite their shared genetic burden. This could be invaluable knowledge for informing prevention strategies.

# 4. Discussion

What we have presented here is a relatively small scale example of the application of GFA on data from 43 MZ twin pairs. We acknowledge that this twin data is quite unique and so wish to stress that the method presented can also offer insights from non-twin data where samples are simply the measurements from individuals. In this case, the insights would be of a different form, for example 'individuals of a certain genotype at given SNPs have increased consumption of $x$, $y$ and $z$ and an increase in cytokines $a$, $b$ and $c$ and decreased methylation at CpGs $l$, $m$ and $n$'. Of course, beyond showing which variables tend to co-occur, it is difficult to know whether or how these are obesity-related without an association to the clinical variables. Hence, in this case, components including clinical variables are the most useful.

In terms of machine learning methodologies, there are two interesting future directions to explore. First, it will be interesting to explore multi-tensor factorizations such as [96] to analyse the twin-pair datasets. Tensor factorizations capture multi-way structures in the data and may reveal previously unseen patterns. In addition, it would be interesting to explore the possibility of nonlinear dependencies across the datasets with kernelized matrix factorization-based approaches [97]. As a further direction, machine learning innovations that allow exploration of new use-cases could be investigated. For example, innovations that make it easy to handle massively high-dimensional datasets such as [98] could enhance the applicability of GFA further. In addition, it could be useful to incorporate prior biological knowledge, for example in the form of pathways or functionally linked networks, to supplement the model's learning process.

There is a huge temptation to throw all possible types of data at methods such as GFA to see what novel associations can be found. We next discuss what other types of data we could include, when available.

It has previously been shown that a wide range of unfavourable alterations in the serum metabolome are associated with abdominal obesity, insulin resistance and low-grade inflammation [99] and so this would be very useful to include in future analyses. Also, adding transcriptomics profiles could add clues to how genotype, DNA methylation and gene expression inter-relate [100].

There is growing evidence that the gut microbiome plays vital roles in health and disease [101,102] and in particular that gut microbiota contribute to the regulation of adiposity and are a mediator of dietary impact on the host metabolic status, although some contradictory evidence also exists [103]. Sonnenburg & Bäckhed [104] have recently reviewed evidence of how the gut microbiota can alter extraction of energy from food, generation of metabolic products, such as short-chain fatty acids, and storage of calories. Despite the likely complexity of processes involved, the broad picture seems to suggest that obesity is associated with a reduced diversity of gut microbiota [105], which would mirror the findings from macroecology, suggesting that biodiversity within an ecosystem can serve as a measure of stability and robustness [106]. It has also been suggested that host genetics may influence the presence of certain microbiota [15]. If microbiome data were available, it would be useful to include in the GFA analysis as an additional variable indicative of the level of microbiome diversity or even a whole matrix containing microbiome or microbial taxa level data. It may also be advisable to include all putative genetic determinants of gut microbiota in the analysis. These involve genes related to diet, metabolism, olfaction and immunity [15]. The number of studies into the effect of the microbiome on health is steadily increasing [107], with new methods emerging to measure its composition [108], and so we envisage many opportunities for incorporating such data into the methods in the future.

With dietary data, it is known that the unreliability of food questionnaires poses a major challenge. In particular, obese individuals tend to misreport in their food and physical activity diaries, and we have shown in [109] that the obese twins of MZ discordant pairs over-report their physical activity, and under-report their food intake. In the current study, this does not pose a major problem because we take the difference values of variables, and this would in fact result in fewer associations being discovered with these self-reported variables. Also, the presence of these self-reported variables does not affect the other associations found by the GFA method. However, it is important to note that there are many areas in obesity research that rely partially or totally on self-report data, because objective measures either do not exist or they do not measure all the desired aspects. For example, as reviewed by Sievänen & Kujala [110], accelerometer data provide seemingly objective measures of physical

activity, but come with limitations. In a similar vein, current and very recent smoking exposure can be fairly reliably measured with biomarkers such as cotinine (past few days) or methylation data (perhaps some months), but a lifetime history of smoking cannot be objectively measured. We think it is highly important to include these self-reported measures in this study as physical activity and eating habits are central to the etiology of obesity. Also, their inclusion is intended to give a taste of the sorts of novel associations that can be achieved with ever greater confidence using the GFA method as the reliability of data improves. For example, there have been recent advances in measuring dietary intake via a urine test [111] or using biomarkers in the blood [112]. Such metabolic profiles could be input into the method to offer more reliable links between diet and obesity.

The latest nutrition research should be considered and the breadth and resolution of dietary information adjusted accordingly in light of promising new associations. Also the role of artificial sweeteners, probiotics and emulsifiers could be evaluated if estimates of these levels were included in the data collection [101,113–115]. Finally, as some dietary compounds have been shown to influence the epigenome, inflammation and microbiota [116–119], it could be enlightening to include estimated levels of such compounds, e.g. resveratrol and quercetin, and some of their putative molecular targets, such as HDACs and NFkappaB, into the data [29] to begin to elucidate a more unified picture of the influence of diet.

There is evidence for an interplay between the stress system and obesity, with increased long-term cortisol levels, as measured in scalp hair, being strongly related to abdominal obesity [120]. Hence adding a hair cortisol concentration measure to the variables could potentially discern stress-related obesity mechanisms. Also, including traditional predictors of obesity, such as parental obesity status and presence of childhood obesity, or the polygenic risk score for obesity as discussed above, might also help to distinguish different classes of obesity and their mechanisms.

It is known that genotype has a large effect on methylation and there are resources available to advise which methylation sites are influenced by the genome and which might be affected by disease-relevant environmental exposures [121]. In the current analysis, we decided to limit the number of SNPs and CpGs and so did not incorporate methylation quantitative trait loci (mQTL) information, which would include the *cis* and *trans* effects in the analysis [100]. Instead, had there been genotype-methylation associations highlighted, we planned to look up mQTL information at a later stage to help gain functional insights. In future studies, however, the mQTL genes and SNPs could also be included to see whether known associations appear and what other variables they are associated with.

# 5. Conclusion

It is now well established that machine learning coupled with good quality data is held as key to future discoveries and advances in almost every imaginable field, with investments in this area a cornerstone of the research and innovation strategies of many companies and governmental funding bodies alike. The current work contributes to efforts to bring machine learning to the field of disease prevention and in particular to obesity research.

In terms of impact of such research, discovering molecular mechanisms of disease can of course guide towards drug discovery directly. Also, discovery of individual risk factors can hope to aid in disease prevention through behavioural change, although it is unclear whether informing people of genotype-based disease risk changes behaviour [122,123], especially for those who are genetically highly susceptible to food-rich environments. However, at a societal level there is hope. If researchers can find incontrovertible evidence that the epidemic of obesity cannot be reversed by individual willpower alone owing to the nature of the molecular mechanisms involved, then it must turn to the governments to create the environment necessary to affect the change. The interactions between the environment and the individual in the development of obesity have been acknowledged and described via a full obesity system map by the UK government in its landmark Foresight report on obesity over a decade ago [7] and there have been calls to 'dust off' the report and embrace even further its recommendations to adopt a 'whole systems approach' to tackling obesity [124]. The more understanding there is of the complex molecular mechanisms involved, the greater the evidence to advocate and inform a societal effort to enable a change at an individual level.

Ethics. This study complies with EU General Data Protection Regulations, and national legislation. It was performed according to the guidelines of the Finnish Advisory Board on Research Integrity (http://www.tenk.fi/en/frontpage). The collection of the Finnish Twin Cohort (FTC) data used in this study was approved by the ethics committees of Helsinki University Central Hospital (113/E3/2001, 249/E5/2001, 346/E0/05, 270/13/03/01/2008).

An informed consent was signed by the FTC subjects before the beginning of the studies. The principles of informed consent in the Declaration of Helsinki were implemented.

Data accessibility. The data from this study has been deposited in the Biobank of the Finnish Institute for Health and Welfare, known as the THL Biobank (Terveyden ja Hyvinvoinnin Laitos, Helsinki, Finland; https://thl.fi/en/web/thl-biobank). THL Biobank holds a large number of high-quality sample collections of great importance to research into the health of the Finnish population. THL Biobank grants access to biological samples and related data to research projects that are of high scientific quality and impact, are ethically conducted, and that correspond with the research areas of THL Biobank (https://thl.fi/en/web/thl-biobank/for-researchers). The application process for accessing samples and related data is described in detail here: https://thl.fi/en/web/thl-biobank/for-researchers/application-process. The dataset used in the present manuscript forms part of the Finnish twin cohort biobank sample set: https://thl.fi/en/web/thl-biobank/for-researchers/sample-collections/twin-study. All academic and commercial researchers from any country can apply to access the data and the final decision on providing access is granted by the Director of the Biobank.

Authors' contributions. M.K. conceived of and coordinated the study and drafted the manuscript; M.K., M.O., S.A.K. and M.A. designed the study; J.K., K.H.P. and M.O. provided and advised on the data used in the analyses; S.B. and T.P. performed the data preprocessing; S.A.K. and M.A. performed the computational analysis using GFA; M.K., K.H.P. and M.O. carried out analysis and interpretation of results; All authors participated in drafting and critically revising the manuscript, gave final approval for publication and agree to be held accountable for the work performed therein.

Competing interests. We have no competing interests.

Funding. J.K. has been supported by the Academy of Finland (grant nos 308248 and 312073). Funding sources for K.H.P. are the Academy of Finland (grant nos 272376, 266286, 314383 and 315035), Finnish Medical Foundation, Finnish Diabetes Research Foundation, Novo Nordisk Foundation, Gyllenberg Foundation, Sigrid Juselius Foundation, Helsinki University Hospital Research Funds, Government Research Funds and University of Helsinki. M.O. has been supported by the Academy of Finland (grant no. 297908), Sigrid Juselius Foundation and University of Helsinki. S.A.K. has been supported by Academy of Finland (grant no. 296516). The funders had no role in planning study design, data collection and analysis, decision to publish, or preparation of the article.

Acknowledgements. M.K. thanks Leonie Bogl for very interesting and useful discussions on the dietary data in the TwinFat study.

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
