## [Reviewer comments · Royal Society Open Science]

Review History

RSOS-200872.R0 (Original submission)

Review form: Reviewer 1

Is the manuscript scientifically sound in its present form?

Yes

Are the interpretations and conclusions justified by the results?

Yes

Is the language acceptable?

Yes

Do you have any ethical concerns with this paper?

No

Have you any concerns about statistical analyses in this paper?

No

Recommendation?

Accept with minor revision (please list in comments)

Comments to the Author(s)

Please see attachment (Appendix A).

Kibble et al. have produced a manuscript entitled "An integrative machine learning approach to discovering multi-level molecular mechanisms of obesity using data from monozygotic twin pairs". The authors have utilised the power of discordant monozygotic (MZ) twins in the setting of body weight with the availability of a multivariate dataset combining clinical, cytokine, genomic, methylation and dietary information. The analysis includes 43 young adult MZ twin pairs (aged 22 – 36) with 25 pairs being weight discordant due to a BMI difference of > 3kg/m². Inclusion of MZ twins has clear biological benefit as this removes one of great issues of genetic confounding that occurs in population studies. The manuscript is focused on the novel methodology of jointly analysing these data using an integrative ML method called Group Factor Analysis (GFA). The authors' manuscript has been transferred to Royal Society Open Science journal which assesses papers only on the basis of their scientific soundness and novelty - not impact - and they have provided a point-by-point response to the previous reviews. My comment re these responses and any further points are detailed below.

Review Responses: Reviewer 1

1. Change the title

- Fair response

2. Rewrite Introduction on importance problem and how GFA fills this gap

- Completed

3. What changes the terms in the energy balance equation are not well understood

- Yes, good to state role of energy imbalance in obesity and that genetics points to central control as fundamental in obesity susceptibility. Have expanded on this in Introduction.

4. Re-order manuscript sections

- Completed

5. More explanation of latent variable and GFA required

- Explanation expanded. Yes, the authors state that similar ML approaches have not been applied twin studies. However, should also include reference to recent review of ML in twins across a range of methodologies from Baird & Hysi (2019)¹.

6. Issues re explanation of GFA

- Explanation fine

7. Include Data Processing Figure

- Provided

8. More clearly signpost that the analysis is performed in the differences between the MZ twins

- Yes, discordance details expanded

9. Interpretation of Heatmaps should be included

- Yes, explanation of heatmap results now included

10. Inappropriate 'sales' terms removed

- Yes, terms removed

11. Issues regarding self-reported Diet and physical activity data

- The authors proved a response to this issue. However, they should also include in the manuscript the points and citations to papers they included (Pietiläinen et al. 2010; Sievänen & Kujala, 2017), regarding this in the paper itself - to show that they understand the difficulty and potential caveats to this analysis and to convey this better for readers.

12. Clearly state new findings

- Yes, key new associations summarised

Review Responses: Reviewer 2

1. Novelty of methods and understanding of pathology needs to be better explained

- Yes, addressed by authors
- 2. Figures substandard – Agree the Figure presentation could be improved
- Authors state they will revise figures if manuscript accepted by journal

Additional Comments

1. Section 1.1 Twin studies – The authors should include the caveat that somatic DNA mutation does occur with age² and will led to minor variation even between MZ twins.
2. How did using a threshold for clinical variables of more than 50% of values missing being left out of the matrices impact on the GFA?
3. The authors need to state more specifically that cell-type heterogeneity is a major issue in these studies and how successfully SVA dealt with this.
4. The authors should discuss any weaknesses of using previously identified EWAS results – such as Wahl et al. that included results from a mixture of populations. Also, could relying on previous DNA methylation results reduce novel insights and limit application?

Minor Point

1. Pg 11 line 17 “Methylation...” – sentence is confusing

1. Baird PN & Hysi P. Twin Registries Moving Forward and Meeting the Future: A Review. *Twin research and human genetics : the official journal of the International Society for Twin Studies* 22, 201-9 (2019).
2. Jaiswal S & Ebert BL. Clonal hematopoiesis in human aging and disease. *Science* 366(2019).

Review form: Reviewer 2

Is the manuscript scientifically sound in its present form?

Yes

Are the interpretations and conclusions justified by the results?

No

Is the language acceptable?

Yes

Do you have any ethical concerns with this paper?

No

Have you any concerns about statistical analyses in this paper?

Yes

Recommendation?

Major revision is needed (please make suggestions in comments)

Comments to the Author(s)

Reading the answers to the referee's reports I appreciate the improvement to the paper, however I am not convinced about the results. It would be great to have a kind of validation. The paper looks to me more a methodological procedure than a scientific paper with robust results. For example, it would be great to compare and use the method for another biological questions showing that this strategy is powerful to catch important biological features.

Decision letter (RSOS-200872.R0)

Dear Dr Kibble,

On behalf of the Editors, we are pleased to inform you that your Manuscript RSOS-200872 "An integrative machine learning approach to discovering multi-level molecular mechanisms of obesity using data from monozygotic twin pairs" has been accepted for publication in Royal Society Open Science subject to minor revision in accordance with the referees' reports. Please find the referees' comments along with any feedback from the Editors below my signature.

Both referees have raised a number of points, and please see the comments of the Associate Editor who would like you to focus on dealing with the minor comments of Reviewer 1 but asks if you could also address some of the points raised by Reviewer 2, although further validation or application of GFA to additional data is not necessary. We invite you to respond to the comments and revise your manuscript. Below the referees' and Editors' comments (where applicable) we provide additional requirements. Final acceptance of your manuscript is dependent on these requirements being met. We provide guidance below to help you prepare your revision.

Please submit your revised manuscript and required files (see below) no later than 7 days from today's (ie 07-Sep-2020) date. Note: the ScholarOne system will 'lock' if submission of the revision is attempted 7 or more days after the deadline. If you do not think you will be able to meet this deadline please contact the editorial office immediately.

on behalf of Professor Andrew Teschendorff (Associate Editor) and Steve Brown (Subject Editor)
openscience@royalsociety.org

Associate Editor Comments to Author (Professor Andrew Teschendorff):

Please revise addressing the remaining points of Reviewer-1. It would be good if the authors could also address some of the points raised by Reviewer-2, although further validation or application of GFA to additional data is not necessary.

Reviewer comments to Author:

Reviewer: 1

Comments to the Author(s)

Please see attachment

Kibble et al. have produced a manuscript entitled “An integrative machine learning approach to discovering multi-level molecular mechanisms of obesity using data from monozygotic twin pairs”. The authors have utilised the power of discordant monozygotic (MZ) twins in the setting of body weight with the availability of a multivariate dataset combining clinical, cytokine, genomic, methylation and dietary information. The analysis includes 43 young adult MZ twin pairs (aged 22 – 36) with 25 pairs being weight discordant due to a BMI difference of > 3kg/m². Inclusion of MZ twins has clear biological benefit as this removes one of great issues of genetic confounding that occurs in population studies. The manuscript is focused on the novel methodology of jointly analysing these data using an integrative ML method called Group Factor Analysis (GFA). The authors’ manuscript has been transferred to Royal Society Open Science journal which assesses papers only on the basis of their scientific soundness and novelty - not impact - and they have provided a point-by-point response to the previous reviews. My comment re these responses and any further points are detailed below.

Review Responses: Reviewer 1

1. Change the title

- Fair response

2. Rewrite Introduction on importance problem and how GFA fills this gap

- Completed

3. What changes the terms in the energy balance equation are not well understood

- Yes, good to state role of energy imbalance in obesity and that genetics points to central control as fundamental in obesity susceptibility. Have expanded on this in Introduction.

4. Re-order manuscript sections

- Completed

5. More explanation of latent variable and GFA required

- Explanation expanded. Yes, the authors state that similar ML approaches have not been applied twin studies. However, should also include reference to recent review of ML in twins across a range of methodologies from Baird & Hysi (2019)¹.

6. Issues re explanation of GFA

- Explanation fine

7. Include Data Processing Figure

- Provided

8. More clearly signpost that the analysis is performed in the differences between the MZ twins

- Yes, discordance details expanded

9. Interpretation of Heatmaps should be included

- Yes, explanation of heatmap results now included

10. Inappropriate ‘sales’ terms removed

- Yes, terms removed

11. Issues regarding self-reported Diet and physical activity data

- The authors proved a response to this issue. However, they should also include in the manuscript the points and citations to papers they included (Pietiläinen et al. 2010; Sievänen & Kujala, 2017), regarding this in the paper itself - to show that they understand the difficulty and potential caveats to this analysis and to convey this better for readers.

12. Clearly state new findings

- Yes, key new associations summarised

Review Responses: Reviewer 2

1. Novelty of methods and understanding of pathology needs to be better explained
- Yes, addressed by authors
2. Figures substandard – Agree the Figure presentation could be improved
- Authors state they will revise figures if manuscript accepted by journal

Additional Comments

1. Section 1.1 Twin studies – The authors should include the caveat that somatic DNA mutation does occur with age² and will led to minor variation even between MZ twins.
2. How did using a threshold for clinical variables of more than 50% of values missing being left out of the matrices impact on the GFA?
3. The authors need to state more specifically that cell-type heterogeneity is a major issue in these studies and how successfully SVA dealt with this.
4. The authors should discuss any weaknesses of using previously identified EWAS results – such as Wahl et al. that included results from a mixture of populations. Also, could relying on previous DNA methylation results reduce novel insights and limit application?

Minor Point

1. Pg 11 line 17 “Methylation...” – sentence is confusing

1. Baird PN & Hysi P. Twin Registries Moving Forward and Meeting the Future: A Review. *Twin research and human genetics : the official journal of the International Society for Twin Studies* 22, 201-9 (2019).
2. Jaiswal S & Ebert BL. Clonal hematopoiesis in human aging and disease. *Science* 366(2019).

Reviewer: 2

Comments to the Author(s)

Reading the answers to the referee's reports I appreciate the improvement to the paper, however I am not convinced about the results. It would be great to have a kind of validation. The paper looks to me more a methodological procedure than a scientific paper with robust results. For example, it would be great to compare and use the method for another biological questions showing that this strategy is powerful to catch important biological features.

===PREPARING YOUR MANUSCRIPT===

- one version identifying all the changes that have been made (for instance, in coloured highlight, in bold text, or tracked changes);
- a 'clean' version of the new manuscript that incorporates the changes made, but does not highlight them.

This version will be used for typesetting.

While not essential, it will speed up the preparation of your manuscript proof if you format your references/bibliography in Vancouver style (please see

<https://royalsociety.org/journals/authors/author-guidelines/#formatting>). You should include DOIs for as many of the references as possible.

===PREPARING YOUR REVISION IN SCHOLARONE===

Author's Response to Decision Letter for (RSOS-200872.R0)

See Appendix B.

Decision letter (RSOS-200872.R1)

Dear Dr Kibble,

It is a pleasure to accept your manuscript entitled "An integrative machine learning approach to discovering multi-level molecular mechanisms of obesity using data from monozygotic twin pairs" in its current form for publication in Royal Society Open Science.

Please ensure that you send to the editorial office the updated emails of both of the following co-authors, as their email addresses are currently marked as invalid by our system:

1. suleiman.khan@helsinki.fi
2. muhammad.ammad-ud-din@helsinki.fi

on behalf of Professor Andrew Teschendorff (Associate Editor) and Steve Brown (Subject Editor)
openscience@royalsociety.org

Appendix A

Kibble et al. have produced a manuscript entitled “An integrative machine learning approach to discovering multi-level molecular mechanisms of obesity using data from monozygotic twin pairs”. The authors have utilised the power of discordant monozygotic (MZ) twins in the setting of body weight with the availability of a multivariate dataset combining clinical, cytokine, genomic, methylation and dietary information. The analysis includes 43 young adult MZ twin pairs (aged 22 – 36) with 25 pairs being weight discordant due to a BMI difference of $> 3\text{kg/m}^2$. Inclusion of MZ twins has clear biological benefit as this removes one of great issues of genetic confounding that occurs in population studies. The manuscript is focused on the novel methodology of jointly analysing these data using an integrative ML method called Group Factor Analysis (GFA). The authors’ manuscript has been transferred to Royal Society Open Science journal which assesses papers only on the basis of their scientific soundness and novelty - not impact - and they have provided a point-by-point response to the previous reviews. My comment re these responses and any further points are detailed below.

Review Responses: Reviewer 1

1. Change the title
 - Fair response
2. Rewrite Introduction on importance problem and how GFA fills this gap
 - Completed
3. What changes the terms in the energy balance equation are not well understood
 - Yes, good to state role of energy imbalance in obesity and that genetics points to central control as fundamental in obesity susceptibility. Have expanded on this in Introduction.
4. Re-order manuscript sections
 - Completed
5. More explanation of latent variable and GFA required
 - Explanation expanded. Yes, the authors state that similar ML approaches have not been applied twin studies. However, should also include reference to recent review of ML in twins across a range of methodologies from Baird & Hysi (2019)¹.
6. Issues re explanation of GFA
 - Explanation fine
7. Include Data Processing Figure
 - Provided
8. More clearly signpost that the analysis is performed in the differences between the MZ twins
 - Yes, discordance details expanded
9. Interpretation of Heatmaps should be included
 - Yes, explanation of heatmap results now included
10. Inappropriate ‘sales’ terms removed
 - Yes, terms removed
11. Issues regarding self-reported Diet and physical activity data
 - The authors proved a response to this issue. However, they should also include in the manuscript the points and citations to papers they included (Pietiläinen et al. 2010; Sievänen & Kujala, 2017), regarding this in the paper itself - to show that they understand the difficulty and potential caveats to this analysis and to convey this better for readers.
12. Clearly state new findings
 - Yes, key new associations summarised

Review Responses: Reviewer 2

1. Novelty of methods and understanding of pathology needs to be better explained
 - Yes, addressed by authors
2. Figures substandard – Agree the Figure presentation could be improved
 - Authors state they will revise figures if manuscript accepted by journal

Additional Comments

1. Section 1.1 Twin studies – The authors should include the caveat that somatic DNA mutation does occur with age² and will led to minor variation even between MZ twins.
2. How did using a threshold for clinical variables of more than 50% of values missing being left out of the matrices impact on the GFA?
3. The authors need to state more specifically that cell-type heterogeneity is a major issue in these studies and how successfully SVA dealt with this.
4. The authors should discuss any weaknesses of using previously identified EWAS results – such as Wahl et al. that included results from a mixture of populations. Also, could relying on previous DNA methylation results reduce novel insights and limit application?

Minor Point

1. Pg 11 line 17 “Methylation...” – sentence is confusing

1. Baird PN & Hysi P. Twin Registries Moving Forward and Meeting the Future: A Review. *Twin research and human genetics : the official journal of the International Society for Twin Studies* **22**, 201-9 (2019).
2. Jaiswal S & Ebert BL. Clonal hematopoiesis in human aging and disease. *Science* **366**(2019).

Appendix B

UNIVERSITY OF
CAMBRIDGE

Department of Applied Mathematics
and Theoretical Physics

17th September 2020

Professor Andrew Teschendorff (Associate Editor) and
Professor Steve Brown (Subject Editor)
Journal of the Royal Society Open Science

Re: An integrative machine learning approach to discovering multi-level molecular mechanisms of obesity using data from monozygotic twin pairs

Dear Professor Teschendorff and Professor Steve Brown,

Thank you for your reply regarding the submission of our manuscript entitled “An integrative machine learning approach to discovering multi-level molecular mechanisms of obesity using data from monozygotic twin pairs”, and for the reviewer comments. We are delighted to hear that our manuscript has been accepted for publication in Royal Society Open Science subject to minor revisions. Please find copied below our response, including an explanation of the changes that we have made to the manuscript.

Sincerely yours,

Dr Milla Kibble
Department of Applied Mathematics & Theoretical Physics
University of Cambridge
Wilberforce Road
Cambridge CB3 0WA, UK
mmk60@cam.ac.uk
+44 (0)1223 764076

We would like to thank the reviewers for their time in reviewing our manuscript, “An integrative machine learning approach to discovering multi-level molecular mechanisms of obesity using data from monozygotic twin pairs”, and for their thoughtful and very useful comments. We now comment on each individual point.

Response to reviewer 1:

5. More explanation of latent variable and GFA required

- Explanation expanded. Yes, the authors state that similar ML approaches have not been applied twin studies. However, should also include reference to recent review of ML in twins across a range of methodologies from Baird & Hysi (2019)1.

Response:

Thank you for drawing our attention to this review. We have added the reference in section 1.1.

7. Include Data Processing Figure

- Provided

Response:

We have added this as Figure 2.

11. Issues regarding self-reported Diet and physical activity data

– The authors proved a response to this issue. However, they should also include in the manuscript the points and citations to papers they included (Pietiläinen et al. 2010; Sievänen & Kujala, 2017), regarding this in the paper itself - to show that they understand the difficulty and potential caveats to this analysis and to convey this better for readers.

Response:

Thank you. We have now incorporated the text from the response to reviewers into the manuscript itself (section 4 paragraph 6).

Review Responses: Reviewer 2

2. Figures substandard – Agree the Figure presentation could be improved

- Authors state they will revise figures if manuscript accepted by journal

Response:

Thank you. We have revised all of the diagrams, with the aim of improving the clarity and quality of presentation.

Additional Comments

1. Section 1.1 Twin studies – The authors should include the caveat that somatic DNA mutation does occur with age2 and will led to minor variation even between MZ twins.

Response:

We have added this comment and reference in section 2.4.

2. How did using a threshold for clinical variables of more than 50% of values missing being left out of the matrices impact on the GFA?

Response:

This meant that certain clinical variables were not included in the analysis at all. Therefore, had there been a relationship with one of these variables, we could not pick it up. However, the clinical variables matrix was still sufficiently large and comprehensive, even after the columns with a large number of missing variables were removed.

3. The authors need to state more specifically that cell-type heterogeneity is a major issue in these studies and how successfully SVA dealt with this.

Response:

We thank the reviewer for this comment. As pointed out by the reviewer we haven't specifically adjusted for cell-type composition and this might have impacted the methylation values in our data. We did not adjust for cell type composition specifically because we wanted to contain all clinical variation that associate with the obesity phenotype, including very low grade inflammation, which may result in differences in the cell type compositions within the BMI discordant twin pairs. However, to account for technical effects during the experiment, we have used the ComBat function from the SVA package. ComBat removes batch effects across samples and other unwanted variation through parametric empirical Bayesian adjustments. We have added a sentence in Section 2.5 to clarify the above.

4. The authors should discuss any weaknesses of using previously identified EWAS results – such as Wahl et al. that included results from a mixture of populations. Also, could relying on previous DNA methylation results reduce novel insights and limit application?

Response:

It is true that it would be better to be able to include all CpGs in the analysis to pick up more novel insights. However, this would have resulted in high dimensional matrices and introduced a large number of noisy sites in the modelling process. Therefore, to focus the analysis on the most significant findings, we feel that the best option was to pick CpG sites that have been shown to associate with BMI. The Wahl et al study was performed on a bigger dataset of 5,387 individuals and including a mixture of populations would have led to the identification of robust CpGs associated with BMI, which are common across all populations.

Also, as we commented in section 2.4, “it is difficult to draw meaningful or actionable hypotheses from genes for which nothing is known”, and the same is true of CpGs. Therefore, in the study we chose to focus on sites where there is already an association with obesity or obesity related phenotypes, to see if we could discover novel molecular mechanisms involving these.

We have added two sentences to section 2.5 to better explain the above.

Minor Point

1. Pg 11 line 17 “Methylation...” – sentence is confusing

Response:

Thank you for pointing this out. The sentence has now been changed to “DNA methylation has been shown to be both stable and dynamic. Across the human postnatal lifetime, stability in methylation is primarily due to genetic contributions, while environmental exposures contribute to methylation dynamics (Reynolds et al 2020).”

Response to reviewer 2:

Reading the answers to the referee's reports I appreciate the improvement to the paper, however I am not convinced about the results. It would be great to have a kind of validation. The paper looks to me more a methodological procedure than a scientific paper with robust results. For example, it would be great to compare and use the method for another biological questions showing that this strategy is powerful to catch important biological features.

Response:

The associate editor has asked “It would be good if the authors could also address some of the points raised by Reviewer-2, although further validation or application of GFA to additional data is not necessary.”

The GFA method has been used successfully in the area of drug discovery and in particular discovery of compound modes of action, as discussed at the beginning of section 2.7: “GFA has been successfully used for identifying structural properties predictive of drug responses (24), cross-organism toxicogenomics (56) as well as highly accurate drug response predictions (28).” However, we are applying the method here to a very different and quite rare set of biological data. It would indeed be interesting to see if the findings can be replicated in another set of trait discordant twin pairs with comparable large multivariate dataset available. Further, while we do not go as far as to experimentally validate molecular mechanisms, we hope that the associations found here may be explored more in further studies.

Additional changes to the manuscript:

We have updated the reference numbering and changed the style to Vancouver, as requested.

All table and figure captions have been moved from the main body of the text to a separate document.

Please note that the more comprehensive data accessibility statement has been saved on the journal portal form (the shorter version is still in the manuscript).